

# A long-term study on zooplankton in two contrasting cascade reservoirs (Iguaçu River, Brazil): effects of inter-annual, seasonal, and environmental factors

Pablo H.S. Picapedra[1], Cleomar Fernandes[2], Juliana Taborda[1,2], Gilmar Baumgartner[1,2] and Paulo V. Sanches[1,2,3]

[1] Programa de Pós-Graduação em Recursos Pesqueiros e Engenharia de Pesca, Universidade Estadual do Oeste do Paraná, Toledo, Paraná, Brazil

[2] Grupo de Pesquisas em Recursos Pesqueiros e Limnologia/Instituto Neotropical de Pesquisas Ambientais, Toledo, Paraná, Brazil

[3] Programa de Pós-Graduação em Ciências Ambientais, Universidade Estadual do Oeste do Paraná, Toledo, Paraná, Brazil

Corresponding author
Pablo H.S. Picapedra,
pablo_picapedra@hotmail.com

## ABSTRACT

**Background**. In reservoirs, zooplankton strongly interact with the physical and chemical characteristics of water, and this interaction is mainly influenced by climate variation and the different methods used to manage the dam water level. Therefore, the aim of this study was to evaluate how the distinct operating modes of two cascade reservoirs affected the richness, abundance, and composition of zooplankton, both spatially (intra and inter-reservoirs) and temporally (annual and seasonal). In this study, the upstream reservoir (Salto Santiago) operates using the storage method, with a water retention time (WRT) of 51 days, whereas the downstream reservoir (Salto Osório) operates using the run-of-river method, with a WRT of 16 days.

**Methods**. Zooplankton samples were collected for 16 consecutive years from the two reservoirs located on the Iguaçu River, Brazil. A total of 720 samples were collected. Four-way ANOVAs were used to determine the differences in richness and abundance of the zooplankton among years, periods, reservoirs, and environments. Multidimensional non-metric scaling (NMDS) and an analysis of similarities (ANOSIM) were used to describe similarity patterns in species composition. Finally, a canonical correspondence analysis (CCA) was used to select the environmental predictors that best explained the variation in zooplankton abundance data.

**Results**. We identified a total of 115 taxa in this study, and rotifers were the richest group. In contrast, the copepods were the most abundant. The four-way ANOVA results showed significant differences in the species richness and abundance of the zooplankton among years, periods, reservoirs, and environments. The NMDS ordination and ANOSIM test indicated that the largest differences in zooplankton species composition were annual and seasonal differences. Finally, the CCA showed that these differences were mainly associated with changes in water transparency, temperature, and the chlorophyll *a*, phosphorus, and total dissolved solids concentrations.

**Discussion**. Inter-annual changes in zooplankton species composition showed that over time, large filters-feeders (e.g., large daphinids and calanoid copepods) were replaced by small cladocerans (e.g., bosminids) and generalist rotifers. The highest species richness was associated with the fluvial environment, whereas the highest

abundance was associated with the transitional and lacustrine reservoir environments. Variations in water temperature, nutrients, and food availability explained the annual and seasonal changes in community structure, whereas variations in the water flow characteristics of the environments explained the longitudinal changes in the richness and abundance of zooplankton in reservoirs. The differences in zooplankton structure between the two reservoirs can be explained by the functional differences between the two systems, such as their WRTs and morphometrics.

## INTRODUCTION

Rising air and water temperatures and changes in hydrological conditions caused by alterations in rainfall seasonality or availability are among the major global concerns of our day (*Nobre et al., 2016*; *Rocha et al., 2019*). In some countries, these concerns are even more relevant owing to their large quantity and diversity of aquatic ecosystems. Currently, one of the biggest challenges has been to predict the consequences of climate change on reservoir systems (*Paerl & Paul, 2012*). Reservoirs are inseparable components of most rivers in South America (*Agostinho et al., 2016*). These engineering works have become increasingly prolific in major river basins, mainly owing to the natural conditions of their predominantly free-flowing rivers, which mean that they have high hydroelectrical potential (*Cella-Ribeiro et al., 2017*). However, a dam significantly alters the riverine ecosystem (*Silva et al., 2014*; *Oliveira et al., 2018*) because the reservoir blocks the free flow of the river and creates a semi-lentic or lentic habitat (*Baxter, 1977*). Important factors, such as the quantity and quality of the water, habitats, nutrient and sediment transport, and water retention time (WRT) can dramatically change (*Baumgartner, Baumgartner & Gomes, 2017*; *Loken et al., 2018*). Thus, either directly or indirectly, artificial variations in water level can affect all aquatic organisms, from primary producers (phytoplankton) to consumers (zooplankton, macroinvertebrates, and fish) (*Baumgartner et al., 2019*).

Therefore, the environmental heterogeneity of physical and chemical characteristics and the type of dam operation are the main drivers of biotic structure (*Vinebrooke et al., 2004*). The operation of dams basically follows two patterns: storage and run-of-river (*Nogueira, Perbiche-Neves & Naliato, 2012*). Storage reservoirs have lentic characteristics and a long WRT, which means that changes occur in the variability of the natural flow regime, as well as associated characteristics, such as the magnitude, frequency, duration, and time and rate of this variability (*Biggs, Nikora & Snelder, 2005*). However, run-of-river reservoirs have semi-lotic characteristics and may accumulate a limited amount or no water and have a short WRT (*Perbiche-Neves & Nogueira, 2010*; *Tang & Cao, 2018*). These differences in hydrological conditions and limnological characteristics may lead to different zooplankton communities throughout the environments of these two types of reservoirs. In addition, seasonal influences may be more or less pronounced according to the operating mode of the

reservoir (*Sartori et al., 2009*; *McManamay et al., 2016*). Fast responses to climate change are expected in systems where lotic conditions predominate, unlike reservoirs operated as lentic environments (*Perbiche-Neves & Nogueira, 2013*).

Although the importance of longitudinal spatial compartments on zooplankton dynamics in reservoirs is understood (*Marzolf, 1990*; *Nogueira, 2001*), few studies have clarified the mechanisms responsible for zooplankton community variations in reservoirs with different cascaded operating modes. Some studies have highlighted differences in water flow regime, depth, particle deposition, transparency, and phytoplankton composition as being responsible for the changes in zooplankton community structure between different types of reservoirs (*Nogueira, Oliveira & Britto, 2008*; *Perbiche-Neves & Nogueira, 2010*; *Okuku et al., 2016*). However, in most cases, these studies were focused on a particular group (e.g., copepods) and many lacked long-term research that would have allowed broader conclusions to be made. Understanding how each mode of operation influences zooplankton communities in associated reservoirs can provide a scientific basis for assessing the effects of climate change and anthropogenic impacts on reservoir ecosystems.

An important aspect of this study is that we analyze the major zooplankton groups (rotifers, cladocerans, and copepods) together; an approach that incorporates an important component: the community composition which results from biotic interactions between different zooplankton groups (*Eskinazi-Sant'Anna et al., 2013*; *Beaver et al., 2018*). Therefore, zooplankton communities have potential value as indicators of various system properties, including trophic conditions (*García-Chicote, Armengol & Rojo, 2018*; *Perbiche-Neves et al., 2019*). Within this context, the aim of this study was to analyze, over 16 consecutive years, the variability in the zooplankton communities of two cascade reservoirs with different operational modes (storage and run-of-river) located on the Iguaçu River. To achieve these objectives, we correlated various environmental variables and the trophic state indices of the two reservoirs with their zooplankton community structures (richness, abundance, and composition).

It was assumed that the reservoirs and their compartments possessed intrinsic differences that determined their physical and chemical characteristics and zooplankton structure. We tested the hypothesis that the smaller WRT of the run-of-river-operated reservoir was a limiting factor on zooplankton development, and that the species richness and abundance were negatively affected by the increased water flow and decreased trophic conditions downstream of the storage reservoir. Furthermore, the compartmentalization of the storage reservoir was dependent on its more resilient physical characteristics, such as its higher WRT, reservoir size, and depth. Therefore, we aimed to answer the following questions: (i) Does the upstream reservoir influence the characterization of the zooplankton and the limnological conditions of the downstream reservoir? (ii) Are the Salto Santiago and Salto Osório reservoirs independent systems? (iii) What are the spatial and temporal dynamics of the zooplankton in the reservoirs?

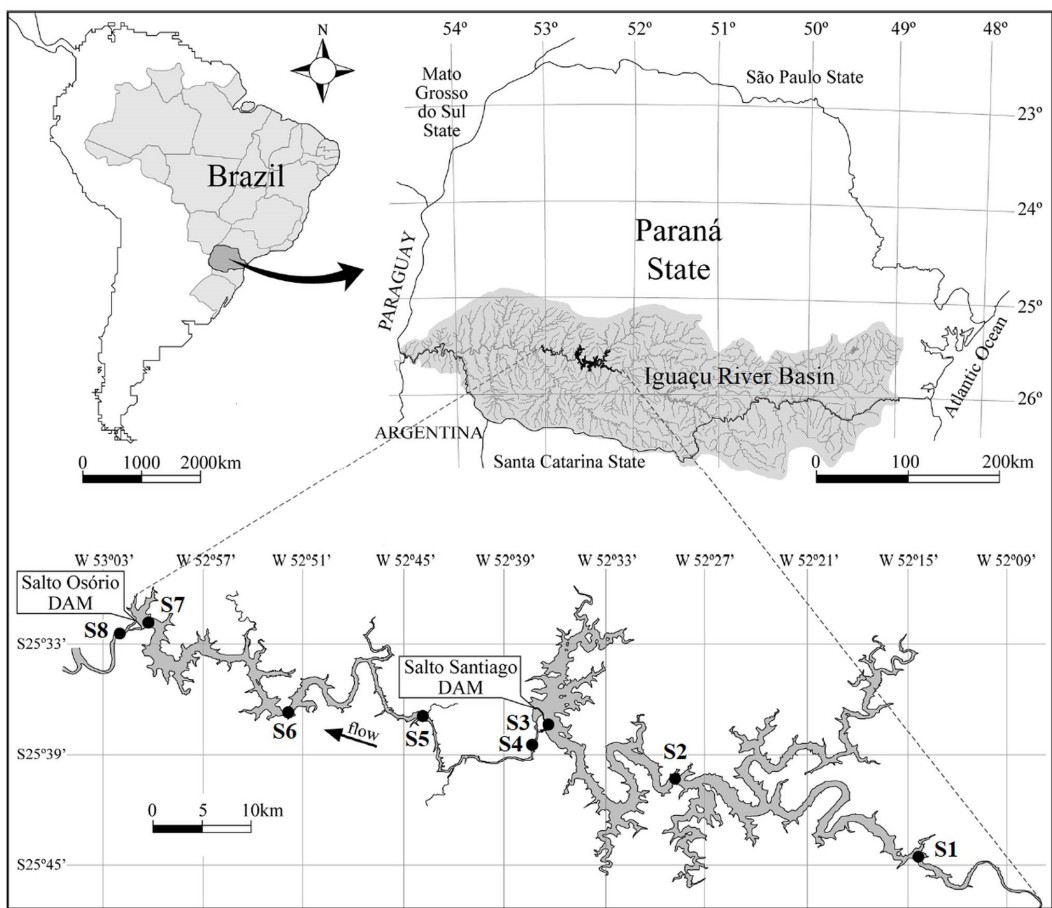

**Figure 1** **Map of the Salto Santiago and Salto Osório reservoirs, and the sampling sites in the Iguaçu River, Brazil.** Environments: fluvial (S1 and S5), transitional (S2 and S6), lacustrine (S3 and S7), and downstream (S4 and S8).

## MATERIALS & METHODS

### Site selection

The Iguaçu River Basin covers an area of approximately 54,820.4 km² across the State of Paraná in Brazil (*Sema-Secretaria do Meio Ambiente, 2010*). The Iguaçu River is a tributary of the Paraná River resulting from the junction of the Iraí and Atuba rivers. The river course follows an east/west direction, with one stretch serving as a natural border between the states of Paraná and Santa Catarina. Another stretch forms the border between Brazil and Argentina. Finally, the Iguaçu River flows into the Paraná River, a few kilometers below the Iguaçu Falls.

This study was conducted at two cascade reservoirs located in the Iguaçu River Basin, southern Brazil (Fig. 1). The Salto Santiago reservoir (storage system) was completed in 1979 and has a watershed area of 43,330 km² and a reservoir area of 208 km². The average annual flow is 902 m³ s, with a cumulative useful volume of 4,094 hm³ and a WRT of 51 days. The sampling sites in the influence area of the reservoir were named S1, S2, S3,

and S4. The Salto Osório reservoir (run-of-river system) was completed in 1975 and has a watershed area of 45,800 km$^2$ and a reservoir area of 51 km$^2$. The average annual flow is 937 m$^3$ s, with a cumulative useful volume of 403 hm$^3$ and a WRT of 16 days. The sampling sites in the influence area of the reservoir were named S5, S6, S7, and S8.

Bimonthly sampling was performed between July 2003 and May 2018, and included the winter (May, July, and September) and summer (November, January, and March) periods. The sampling sites were based on water flow characteristics (*Thornton, Kimmel & Payne, 1990*), and four environments were identified. These were: (i) the fluvial zone: a lotic environment located upstream of the dam that has a high water flow and is not directly influenced by the reservoir (S1 and S5); (ii) the transitional zone: a lotic/lentic transition environment that is located between the dam and some tributaries. It has moderate water movement that allows the formation of backwater areas (S2 and S6); (iii) the lacustrine zone: an environment located near the dam and has standing or slow water (S3 and S7); and (iv) the downstream zone: a lotic environment downstream of the dam that has a high water flow and is directly influenced by the reservoir (S4 and S8) (Fig. 1).

## Sampling and quantification of environmental variables

Dissolved oxygen (mg L$^{-1}$; portable oximeter, YSI 550A), electrical conductivity ($\mu$S cm$^{-1}$; portable conductivity meter, Digimed$^{\circledR}$ DM-3P), pH (portable pH meter, Digimed$^{\circledR}$ DM-2P), water temperature (C$^{\circ}$; mercury bulb thermometer), and turbidity (NTU; portable turbidity meter, LaMotte$^{\circledR}$ 2020i) were measured at the water subsurface in each environment. We also collected water samples from the subsurface with a Van Dorn bottle (2.5 L capacity) to determine other variables: total phosphorus (mg L$^{-1}$; *Apha-American Public Health Association, 2005*), total nitrogen (mg L$^{-1}$; *Mackereth, Heron & Talling, 1978*), total dissolved solids (mg L$^{-1}$; *Wetzel & Likens, 2000*), and chlorophyll *a* ($\mu$g L$^{-1}$; *Golterman, Clyno & Ohnstad, 1978*). Moreover, we determined the maximum depth (Z$_{max}$, (m)) and the transparency of the water column using a Secchi disk (Z$_{SD}$, (m)). In addition, the Agência Nacional das Águas (ANA) provided water flow values (m$^{-3}$ s) for the lacustrine zone of each reservoir from the hydrological stations located in the dams. Finally, the ANA also provided precipitation values (mm) for the region through the Saudade do Iguaçu weather station.

## Sampling, identification, and quantification of the zooplankton

Zooplankton samples (720 in total) were taken at each site using a conical plankton net with a mesh size of 68 $\mu$m. We filtered 200 liters of subsurface water per sample using a motor-pump. The collected material was placed in polyethylene bottles (500 mL), labeled, and fixed in 4% formaldehyde buffered with sodium borate (Na$_3$BO$_3$).

Sedgewick-Rafter chambers were used to quantify the zooplankton. Standardized volume (50 mL) aliquots were removed from the samples using a Hensen-Stempell pipette (2.5 mL) and used to count the zooplankton. At least 50 rotifers, cladocerans, young forms (nauplii and copepodites), and adult copepod individuals were counted (*Bottrell et al., 1976*) under an optical microscope with a magnification range of ×10 to ×100. Density was expressed in terms of individuals per m$^{-3}$. The species were identified using

*Koste (1978)*, *Reid (1985)*, *Matsumura-Tundisi (1986)*, and *Elmoor-Loureiro (1997)*. The zooplankton samples were deposited in the Grupo de Pesquisas em Recursos Pesqueiros e Limnologia of the Universidade Estadual do Oeste do Paraná, Campus Toledo, Brazil.

## Data analyses

The trophic state index (TSI) was calculated for each sampling site according to *Carlson (1977)*, as modified by *Lamparelli (2004)*, using the chlorophyll *a*, total phosphorus, and water transparency values. These values were correlated with the zooplankton. A four-way analysis of variance (four-way ANOVA) was used, with a significance level of $P < 0.05$, to investigate temporal and spatial changes in zooplankton richness (number of species) and abundance, and the environmental variables. The analyses considered the following factors: year of sampling (2003–18), period (winter and summer), reservoir (SS, Salto Santiago and SO, Salto Osòrio), and environment (fluvial, transitional, lacustrine, and downstream), as well as the interactions between them (*Sokal & Rohlf, 1991*). Normality and homoscedasticity (homogeneity of variance) were initially verified using the Shapiro–Wilk and Levene tests, respectively. When the ANOVA was significant, we used Tukey's post hoc test to investigate the differences between pairs.

We performed a multidimensional non-metric scaling (NMDS) analysis using a Bray–Curtis distance matrix in order to determine the similarities in the zooplankton communities based on density (*Oksanen et al., 2016*). Young copepod forms (nauplii and copepodites) were not included in the analysis because they generally represent more than one species. The positions of the samples in relation to the year of sampling, period, and environment were used as symbol factors for each reservoir. Stress values were used to determine the number of dimensions. A two-dimensional solution was chosen because there were only small stress value changes in the sequential dimensions. Subsequently, an analysis of similarity (ANOSIM) was used to verify the statistical significance of the groups identified by the NMDS analysis (*Clarke, 1993*). In addition, a similarity percentage test (SIMPER) was used to evaluate the contribution of each species to the group separation found by the NMDS analysis (*Clarke & Warwick, 2001*).

Finally, a canonical correspondence analysis (CCA) was used to select the predictors that best explained the variation in zooplankton composition. We only used the density data for the species that most contributed to community temporal and spatial variation according to the SIMPER test. The environmental variables were submitted to a stepwise forward selection procedure in which the statistical significance of each variable was tested by the Monte Carlo permutation test (999 permutations) with a cutoff point of $P < 0.05$ (*Ter-Braak & Verdonschot, 1995*).

All data (except pH) were log $(x + 1)$ transformed prior to analysis to reduce the influence of outliers. The ANOVA, NMDS, ANOSIM, SIMPER, and CCA analyses were performed by the statistical environment in R version 3.0.2 (*R Core Team, 2015*) and the Vegan R version 2.0–6 package (*Oksanen et al., 2016*).

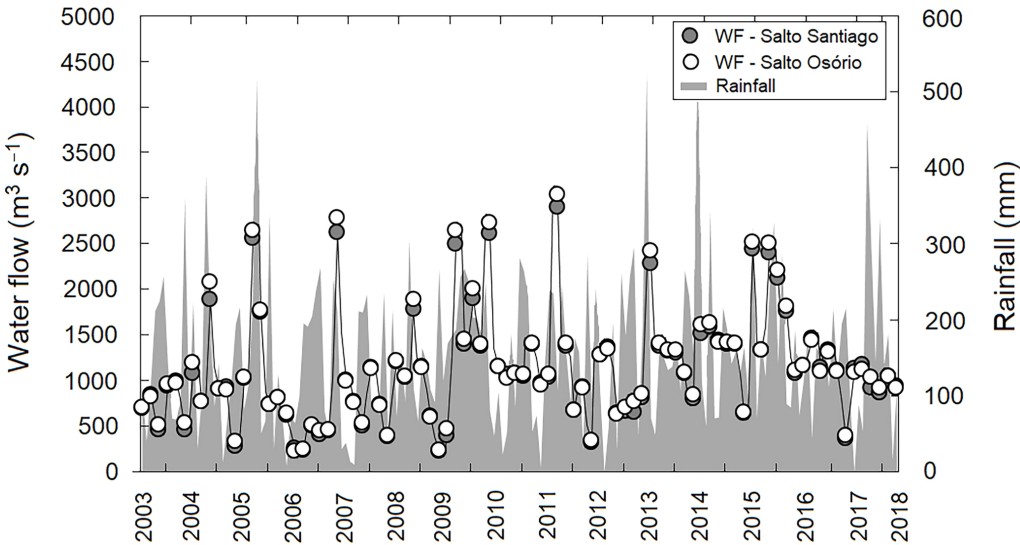

**Figure 2** **Variation in rainfall and water flow (WF) during the study period for the region.** Region comprised of the Salto Santiago and Salto Osório reservoirs, Iguaçu River, Brazil. Data are the total accumulated rainfall and mean water flow values in the reservoirs during the sampling months.

## RESULTS

### Environmental conditions

A lack of seasonality was observed in the region when the average precipitation values during the study period were evaluated ($P > 0.05$). This indicates that throughout the study period there was no pattern in the rainfall regime, with dry and wet winter and summer periods. In contrast, annual differences in rainfall values were observed ($P < 0.05$), with cumulative rainfall peaks (>450 mm) recorded in November 2005, July 2013, and November 2017. May 2011, and July 2012 and 2017, showed the lowest cumulative rainfall (<6.0 mm) for the study region. Consequently, this large annual variability in precipitation had a direct effect on reservoir water flow regulation between years ($P < 0.05$). The highest water flow values for both reservoirs were recorded in September 2011 (SS: 2,894 $m^{-3}$ s and SO: 3,036 $m^{-3}$ s) (Fig. 2).

In general, the reservoirs were classified as mesotrophic according to the TSI classification. The upstream and downstream environments of the dams revealed that there was an increase in average TSI values between 2007–11 and 2013–14. However, after 2014 a decrease in TSI values was noticeable, which showed that oligotrophication had occurred (Fig. 3). The TSI values were higher for environments influenced by the SS reservoir. In addition, the TSI only showed an environmental gradient for the SS reservoir, which decreased between the fluvial and lacustrine zones. In contrast the SO reservoir results showed that the environment was homogeneous.

All the environmental variables showed heterogeneity over the years ($P < 0.05$) (Table 1). A table listing the values for each measured variable and the complete four-way ANOVA results is shown in Data S1 and Table S1, respectively. The WT, DO, and TP values were

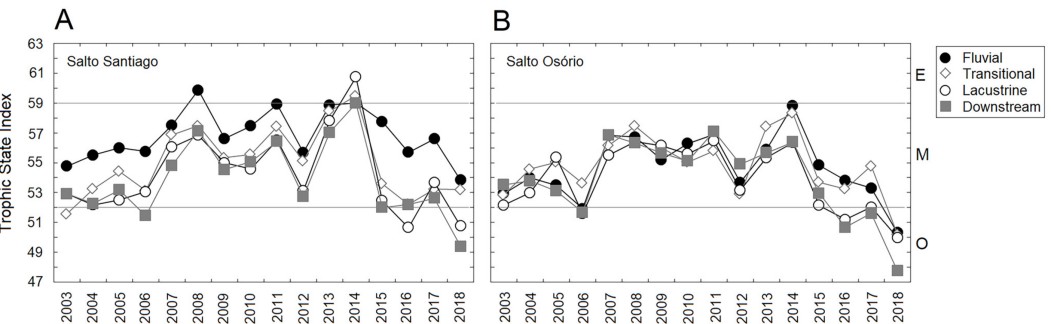

**Figure 3 Trophic state indexes (TSIs).** Mean values of the TSIs calculated for the environments influenced by the (A) Salto Santiago and (B) Salto Osório reservoirs. Horizontal lines indicate the ranges for each trophic state: oligotrophic (O), mesotrophic (M), and eutrophic (E).

**Table 1 Hydrological data.** Mean values, coefficients of variation (CV in %) and ANOVA results for the parameters in each environment and for seasonality (winter and summer periods) in the study region.

| | Chl-*a* | DO | $Z_{max}$ | $Z_{SD}$ | Cond | pH | TDS | TN | TP | Turb | WT | TSI |
|---|---|---|---|---|---|---|---|---|---|---|---|---|
| S1 | 1.6 | 7.3 | 38.2 | 1.6 | 42.6 | 6.9 | 5.5 | 0.67 | 0.09 | 9.0 | 21.7 | 57.0 |
| | 121% | 21% | 18% | 33% | 26% | 11% | 469% | 131% | 96% | 64% | 15% | 5% |
| S2 | 2.2 | 7.7 | 62.1 | 2.3 | 42.3 | 7.2 | 2.9 | 0.61 | 0.08 | 6.1 | 22.9 | 55.2 |
| | 176% | 19% | 19% | 35% | 27% | 12% | 141% | 140% | 106% | 97% | 17% | 7% |
| S3 | 2.4 | 7.6 | 64.9 | 2.6 | 42.4 | 7.1 | 4.4 | 0.59 | 0.08 | 7.4 | 23.3 | 54.5 |
| | 254% | 16% | 13% | 35% | 26% | 11% | 267% | 115% | 105% | 312% | 15% | 8% |
| S4 | 1.3 | 7.3 | 7.9 | 2.5 | 41.6 | 6.9 | 6.0 | 0.64 | 0.08 | 5.9 | 21.1 | 54.1 |
| | 166% | 23% | 6% | 36% | 29% | 8% | 375% | 123% | 107% | 105% | 13% | 6% |
| S5 | 1.2 | 7.4 | 33.1 | 2.3 | 43.1 | 7.0 | 4.5 | 0.64 | 0.08 | 7.9 | 21.1 | 54.9 |
| | 101% | 23% | 17% | 37% | 27% | 8% | 359% | 122% | 100% | 165% | 14% | 6% |
| S6 | 1.4 | 7.6 | 34.2 | 2.3 | 43.0 | 7.0 | 2.6 | 0.54 | 0.08 | 7.1 | 22.4 | 55.0 |
| | 120% | 23% | 19% | 38% | 30% | 9% | 120% | 120% | 100% | 119% | 15% | 6% |
| S7 | 1.4 | 7.7 | 42.0 | 2.6 | 43.1 | 7.1 | 3.2 | 0.61 | 0.08 | 5.8 | 23.2 | 54.1 |
| | 143% | 19% | 9% | 34% | 27% | 11% | 162% | 108% | 105% | 130% | 15% | 6% |
| S8 | 1.3 | 7.0 | 5.9 | 2.5 | 41.5 | 5.9 | 2.4 | 0.59 | 0.08 | 6.2 | 21.4 | 54.2 |
| | 218% | 23% | 7% | 39% | 28% | 7% | 130% | 116% | 101% | 99% | 13% | 6% |
| Winter | 1.4 | 7.8 | 36.2 | 2.4 | 42.7 | 7.0 | 3.0 | 0.62 | 0.08 | 8.0 | 19.6 | 55.0 |
| | 142% | 20% | 59% | 41% | 32% | 10% | 133% | 107% | 99% | 185% | 10% | 6% |
| Summer | 1.8 | 7.1 | 35.9 | 2.3 | 42.2 | 7.0 | 4.9 | 0.61 | 0.07 | 5.8 | 24.6 | 54.7 |
| | 217% | 20% | 59% | 36% | 21% | 10% | 402% | 136% | 105% | 76% | 10% | 7% |
| *P* values | | | | | | | | | | | | |
| Year | **<0.001** | **<0.001** | **<0.001** | **<0.001** | **<0.001** | **<0.001** | **<0.001** | **<0.001** | **<0.001** | **<0.001** | **0.012** | **<0.001** |
| Period | 0.559 | **<0.001** | 0.363 | 0.473 | 0.526 | 0.221 | 0.623 | 0.149 | **0.027** | 0.279 | **<0.001** | 0.131 |
| Reservoir | **0.026** | 0.406 | **<0.001** | **0.001** | 0.544 | 0.757 | 0.171 | 0.716 | 0.656 | 0.368 | 0.225 | **0.003** |
| Environment | **0.047** | **0.001** | **<0.001** | **<0.001** | 0.394 | **0.008** | 0.461 | 0.609 | 0.914 | **<0.001** | **<0.001** | **<0.001** |

**Notes.**
Bold ANOVA results (*P* values) are significant ($P < 0.05$). Chlorophyll a (Chl-*a*), dissolved oxygen (DO), maximum depth ($Z_{max}$), Secchi disk depth (*ZSD*), electrical conductivity (Cond), total dissolved solids (TDS), total nitrogen (TN), total phosphorus (TP), turbidity (Turb), water temperature (WT), and trophic state index (TSI).
affected by the seasons, with higher WT values in summer, and higher DO and TP values in winter. The environments influenced by the SS reservoir had higher Chl-$a$, $Z_{max}$, and $Z_{SD}$ values. In addition, a gradient was observed for the $Z_{max}$, DO, pH, and WT values, with a gradual increase between the fluvial and lacustrine zones in the two reservoirs. However, there were only gradients for the $Z_{SD}$ and Turb values in SS reservoir, with the $Z_{SD}$ values increasing between the fluvial and lacustrine zones, whereas Turb values decreased. Finally, the Chl-$a$ concentrations were lower downstream compared to the reservoir transitional zones (Table 1).

## General scenario for the zooplankton communities

The zooplankton community was composed of 115 taxa. Rotifera were the richest species group (74 species), followed by Cladocera (26 species) and Copepoda (15 species). There was a larger number of zooplankton taxa in the environments with lotic characteristics (fluvial and downstream zones) for both reservoirs (Table 2). The frequency (number of samples in which a species occurred) of the different taxa did not show any major changes between environments. In addition, some taxa had a relatively high frequency in the study ($\geq$70%), e.g., *Polyarthra* sp., *Bosmina hagmanni*, *Ceriodaphnia cornuta*, *Ceriodaphnia silvestrii*, *Daphnia gessneri*, and *Notodiaptomus henseni*. Young copepod forms (nauplii and copepodits) were more frequent than adult forms (Table 2). The zooplankton community abundances of the two reservoirs were mainly driven by the copepods, which were mainly represented by nauplii and copepodites. However, the most abundant adult species were Calanoids, *Notodiaptomus transitans*, and *N. henseni*. The cladocerans represented the second most abundant group in the study, particularly *C. cornuta* and *B. hagmanni*. *Conochilus* sp. *Asplanchna sieboldii*, and *Polyarthra* sp. were the most abundant rotifer species (Table 2). A table listing the number of taxa found and the abundance values for each zooplankton taxon is shown in Data S2.

## Zooplankton richness and abundance

Significant changes ($P < 0.05$) were observed among the years for total zooplankton species richness and among the different groups in the two reservoirs (Table 3). The results showed that there was a gradual increase in the number of species from 2010, and that rotifers and cladocerans contributed most over the years to the total richness of the zooplankton community (Fig. 4). The total abundance of the zooplankton and the different groups also fluctuated among the years in the two reservoirs (Table 3). The microcrustaceans (copepods and cladocerans) contributed most to community abundance between 2003–12. However, from 2013, the copepods and rotifers were the predominant organisms in the zooplankton (Fig. 4).

There was a small seasonal change in cladoceran richness, with higher values in summer. However, in general, there was no established pattern when all the study years were taken into account (Table 3; Fig. 5). The seasons did affect abundance, with higher values in summer for all groups. However, abundance did not follow a cyclical pattern between the winter and summer periods over the years (Table 3; Fig. 5).

The total species richness, and rotifers and cladocerans species values were higher in the SS reservoir. There were also differences in species richness between the different

**Table 2** Taxonomic composition, frequency of occurrence (%) and mean abundance (in parentheses) of each zooplankton taxa in each environment influenced by the Salto Santiago and Salto Osório reservoirs.

| Taxa | S1 | S2 | S3 | S4 | S5 | S6 | S7 | S8 |
|------|----|----|----|----|----|----|----|----|
| Rotifera | | | | | | | | |
| *Ascomorpha* sp. | 22 (246) | 17 (64) | 11 (11) | 20 (35) | 17 (24) | 20 (228) | 14 (77) | 16 (14) |
| *Asplanchna sieboldii* | 34 (3,313) | 38 (5,258) | 32 (1,943) | 31 (615) | 29 (141) | 30 (293) | 31 (1,211) | 27 (81) |
| *Brachionus calyciflorus* | 18 (404) | 11 (309) | 12 (427) | 11 (337) | 17 (101) | 14 (851) | 12 (602) | 18 (1,070) |
| *Brachionus dolabratus* | 9 (14) | 11 (82) | 16 (138) | 19 (26) | 14 (29) | 10 (19) | 12 (44) | 9 (21) |
| *Brachionus falcatus* | 14 (101) | 17 (34) | 19 (121) | 18 (32) | 21 (27) | 19 (50) | 16 (47) | 14 (39) |
| *Collotheca* sp. | 9 (17) | 9 (18) | 12 (46) | 10 (14) | 13 (28) | 9 (17) | 10 (172) | 10 (3) |
| *Conochilus* sp. | 64 (3,570) | 67 (6,448) | 67 (6,598) | 66 (1,918) | 58 (807) | 61 (3,129) | 59 (2,633) | 51 (1,210) |
| *Euchlanis dilatata* | 31 (881) | 46 (961) | 39 (677) | 48 (1,045) | 40 (855) | 27 (793) | 22 (671) | 14 (464) |
| *Filinia opoliensis* | 24 (707) | 24 (191) | 12 (88) | 20 (56) | 18 (89) | 11 (41) | 11 (32) | 10 (411) |
| *Hexarthra* sp. | 57 (1,864) | 62 (1,462) | 60 (3,002) | 57 (639) | 56 (794) | 49 (420) | 48 (798) | 28 (379) |
| *Kellicottia bostoniensis* | 67 (1,204) | 50 (1,097) | 49 (614) | 62 (713) | 67 (779) | 57 (958) | 38 (477) | 56 (1,134) |
| *Keratella americana* | 33 (107) | 31 (69) | 33 (66) | 34 (42) | 31 (35) | 31 (79) | 23 (29) | 27 (21) |
| *Keratella cochlearis* | 51 (1,717) | 51 (587) | 44 (176) | 46 (132) | 51 (215) | 39 (280) | 44 (177) | 38 (59) |
| *Keratella tropica* | 39 (596) | 27 (311) | 24 (136) | 19 (109) | 31 (470) | 23 (488) | 19 (215) | 14 (127) |
| *Ploesoma truncatum* | 36 (377) | 21 (673) | 17 (94) | 47 (184) | 42 (148) | 23 (52) | 20 (127) | 36 (140) |
| *Polyarthra* sp. | 82 (5,232) | 82 (5,859) | 70 (4,870) | 82 (1,992) | 74 (2,444) | 76 (3,583) | 79 (2,002) | 77 (1,042) |
| *Ptygura* sp. | 11 (82) | 10 (11) | 14 (245) | 16 (44) | 14 (35) | 11 (7) | 10 (11) | 14 (44) |
| *Synchaeta* sp. | 57 (1,925) | 52 (1,920) | 44 (1,080) | 34 (341) | 50 (751) | 44 (798) | 42 (578) | 36 (739) |
| *Trichocerca cylindrica* | 36 (441) | 44 (1,031) | 41 (708) | 54 (703) | 50 (594) | 31 (316) | 49 (698) | 44 (470) |
| *Trichocerca similis* | 14 (51) | 7 (62) | 8 (11) | 10 (14) | 12 (18) | 7 (35) | 7 (8) | 12 (9) |
| Cladocera | | | | | | | | |
| *Bosmina freyi* | 20 (95) | 19 (303) | 21 (205) | 20 (125) | 21 (104) | 21 (279) | 18 (189) | 19 (52) |
| *Bosmina hagmanni* | 90 (3,703) | 98 (5,608) | 98 (7,256) | 94 (2,480) | 93 (2,828) | 93 (4,115) | 90 (3,172) | 92 (2,814) |
| *Bosminopsis deitersi* | 56 (876) | 38 (422) | 29 (116) | 24 (82) | 37 (118) | 29 (376) | 20 (61) | 21 (85) |
| *Ceriodaphnia cornuta* | 87 (5,255) | 92 (7,182) | 88 (6,138) | 92 (2,593) | 86 (3,590) | 81 (4,417) | 93 (5,854) | 86 (2,945) |
| *Ceriodaphnia silvestrii* | 74 (1,947) | 81 (3,734) | 83 (2,535) | 84 (2,203) | 82 (2,198) | 78 (1,780) | 79 (3,611) | 78 (2,249) |
| *Daphnia gessneri* | 76 (1,609) | 87 (2,498) | 83 (1,980) | 87 (889) | 86 (2,183) | 81 (1,605) | 84 (2,359) | 83 (3,770) |
| *Diaphanosoma birgei* | 12 (47) | 19 (216) | 21 (208) | 17 (41) | 9 (110) | 21 (86) | 20 (213) | 16 (41) |
| *Diaphanosoma spinulosum* | 59 (676) | 64 (1,402) | 69 (2,917) | 67 (782) | 58 (476) | 61 (1,266) | 69 (2,166) | 52 (1,382) |
| *Moina minuta* | 68 (2,903) | 72 (2,395) | 72 (2,703) | 64 (1,010) | 69 (2,086) | 63 (1,544) | 64 (2,309) | 59 (1,988) |
| Copepoda | | | | | | | | |
| Nauplii Cyclopoida | 92 (12,376) | 91 (12,127) | 88 (9,239) | 89 (5,787) | 86 (4,589) | 90 (7,109) | 87 (6,868) | 94 (4,960) |
| Nauplii Calanoida | 99 (8,775) | 97 (9,091) | 93 (7,001) | 96 (3,018) | 92 (3,815) | 94 (6,064) | 96 (6,422) | 92 (4,061) |
| Copepodit Cyclopoida | 88 (2,566) | 83 (2,921) | 70 (2,665) | 77 (1,273) | 81 (2,057) | 69 (1,694) | 77 (1,549) | 78 (1,497) |
| Copepodit Calanoida | 97 (11,168) | 94 (10,107) | 97 (14,235) | 98 (6,762) | 99 (8,511) | 99 (12,681) | 100 (15,509) | 100 (11,281) |
| *Acanthocyclops robustus* | 18 (22) | 12 (14) | 12 (13) | 14 (9) | 17 (13) | 13 (13) | 10 (29) | 12 (4) |
| *Argyrodiaptomus furcatus* | 17 (81) | 13 (62) | 14 (47) | 10 (18) | 12 (28) | 14 (76) | 10 (45) | 12 (21) |
| *Mesocyclops meridianus* | 14 (59) | 16 (26) | 13 (15) | 18 (21) | 17 (16) | 16 (37) | 13 (65) | 13 (12) |

**Table 2** (*continued*)

| Taxa | S1 | S2 | S3 | S4 | S5 | S6 | S7 | S8 |
|------|----|----|----|----|----|----|----|----|
| *Notodiaptomus henseni* | 78 (1,630) | 80 (2,320) | 79 (2,086) | 82 (923) | 79 (926) | 80 (2,525) | 87 (2,969) | 76 (1,051) |
| *Notodiaptomus iheringi* | 21 (123) | 18 (122) | 14 (125) | 14 (63) | 16 (66) | 16 (101) | 27 (249) | 18 (67) |
| *Notodiaptomus* sp. | 7 (32) | 11 (17) | 11 (14) | 7 (9) | 9 (29) | 13 (45) | 13 (69) | 11 (15) |
| *Notodiaptomus transitans* | 50 (1,959) | 53 (5,151) | 48 (1,604) | 49 (1,060) | 49 (1,024) | 52 (2,447) | 57 (2,184) | 38 (1,016) |
| *Thermocyclops decipiens* | 60 (987) | 53 (620) | 57 (571) | 56 (391) | 60 (612) | 56 (340) | 54 (480) | 60 (331) |
| *Thermocyclops minutus* | 41 (337) | 37 (237) | 32 (155) | 41 (111) | 31 (217) | 32 (139) | 37 (99) | 29 (24) |
| Total richness | 82 | 76 | 76 | 80 | 86 | 85 | 81 | 88 |

**Notes.**

Environments: fluvial (S1 and S5), transitional (S2 and S6), lacustrine (S3 and S7), and downstream (S4 and S8). Rare species (≤10%) in the study are not included in the table.

**Table 3  Results ($P$-values) of four-way ANOVA for the zooplankton richness and abundance.** Year indicates the sampling years (2003–18); period refers to winter and summer; reservoir refers to Salto Santiago and Salto Osório reservoirs; and environment refers to the four environments (fluvial, transitional, lacustrine, and downstream). Significant differences ($P < 0.05$) are shown in bold; asterisks indicate an interaction between the effects.

| | Richness | | | | Abundance | | | |
|---|---|---|---|---|---|---|---|---|
| Effects | Total | Rotifera | Cladocera | Copepoda | Total | Rotifera | Cladocera | Copepoda |
| Year | **<0.001** | **<0.001** | **<0.001** | **<0.001** | **<0.001** | **<0.001** | **<0.001** | **<0.001** |
| Period | 0.595 | 0.385 | **0.008** | 0.832 | **<0.001** | **<0.001** | **<0.001** | **<0.001** |
| Reservoir | **<0.001** | **<0.001** | **0.033** | 0.586 | **<0.001** | **<0.001** | **0.001** | **0.001** |
| Environment | **0.034** | **<0.001** | 0.471 | 0.195 | **<0.001** | **<0.001** | **<0.001** | **<0.001** |
| Year*Period | **<0.001** | **<0.001** | **0.015** | **0.001** | **0.005** | **<0.001** | **<0.001** | **0.033** |
| Year*Reservoir | 0.424 | 0.159 | 0.674 | 0.965 | **0.016** | 0.438 | 0.224 | **0.045** |
| Period*Reservoir | 0.979 | 0.723 | **0.024** | 0.56 | 0.188 | 0.421 | **0.039** | 0.687 |
| Year*Environment | 0.847 | 0.986 | 0.751 | 0.146 | 0.436 | 0.437 | 0.604 | 0.81 |
| Period*Environment | 0.264 | 0.916 | 0.064 | 0.74 | **0.047** | 0.097 | 0.128 | **0.039** |
| Reservoir*Environment | **0.042** | 0.174 | 0.158 | 0.584 | **0.041** | 0.484 | 0.402 | **0.003** |
| Year*Period*Reservoir | 0.192 | 0.186 | 0.463 | 0.557 | 0.946 | 0.654 | 0.853 | 0.5 |
| Year*Period*Environment | 0.981 | 0.944 | 0.999 | 0.901 | 0.988 | 0.99 | 0.997 | 0.981 |
| Year*Reservoir*Environment | 0.987 | 0.467 | 0.986 | 0.887 | 0.899 | 0.925 | 0.962 | 0.883 |
| Period*Reservoir*Environment | 0.987 | 0.524 | 0.653 | 0.461 | 0.928 | 0.207 | 0.942 | 0.75 |
| Year*Period*Reservoir*Environment | 0.996 | 0.988 | 0.996 | 0.447 | 0.952 | 0.996 | 0.989 | 0.966 |

environments. The fluvial zone had higher total richness values, and the rotifer values were higher in the fluvial-transitional zones. However, there were total species richness differences between the reservoirs. Species richness was homogeneous in the SS reservoir and heterogeneous in the SO reservoir, with higher values for the fluvial zone compared to downstream (Table 3; Fig. 6).

The results showed that the reservoir abundance values for all zooplankton groups were significantly higher in the SS reservoir. In addition, there were significant changes in abundances between the environments (Table 3) because the total abundance and rotifers abundance values upstream of the dams (fluvial, transitional and lacustrine zones), and for microcrustaceans in low flow environments (transitional and lacustrine zones) were higher than in the other zones. There were also differences in copepod abundances and the total abundance of the community between the reservoirs (Table 3). The SS reservoir

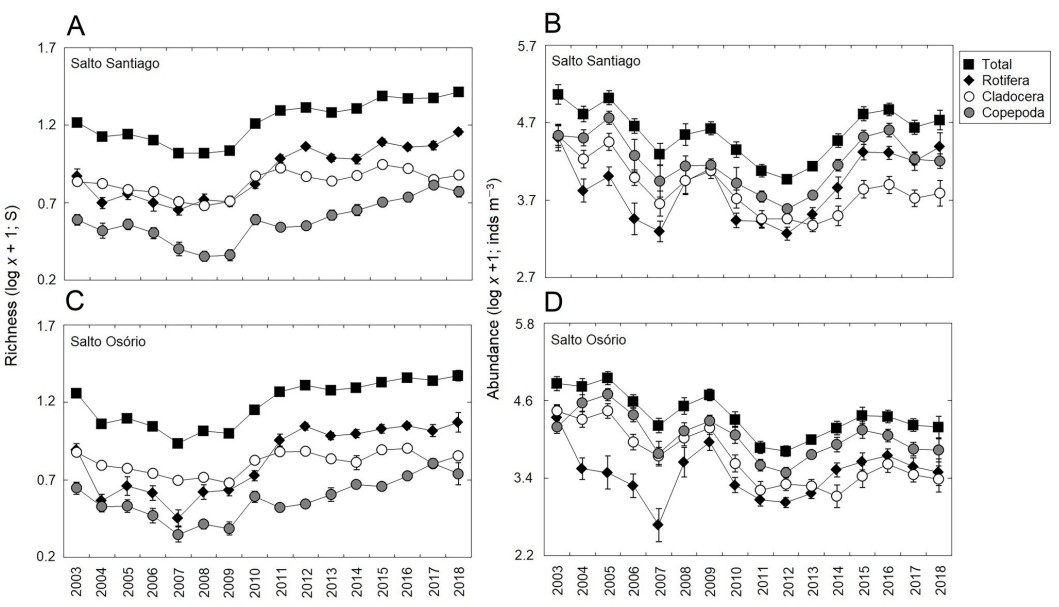

**Figure 4  Inter-annual variations in the richness and abundance of zooplankton.** Data shows the mean (±SE) richness and abundance values for total zooplankton, rotifers, cladocerans, and copepods in the (A, B) Salto Santiago and (C, D) Salto Osório reservoirs.

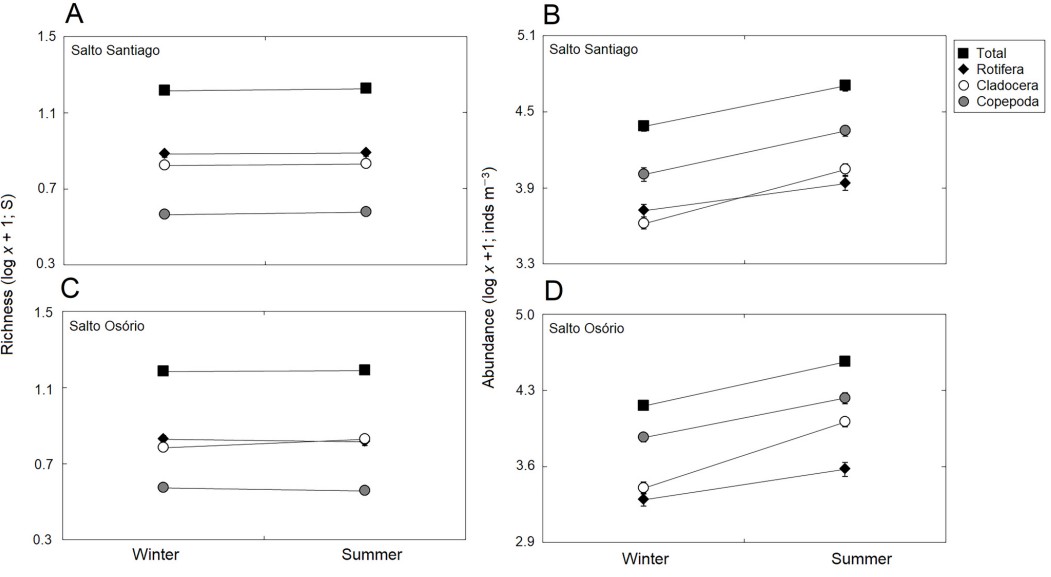

**Figure 5  Seasonal variations in zooplankton richness and abundance.** Data shows the mean (±SE) richness and abundance of total zooplankton, rotifers, cladocerans, and copepods in the (A, B) Salto Santiago and (C, D) Salto Osório reservoirs.

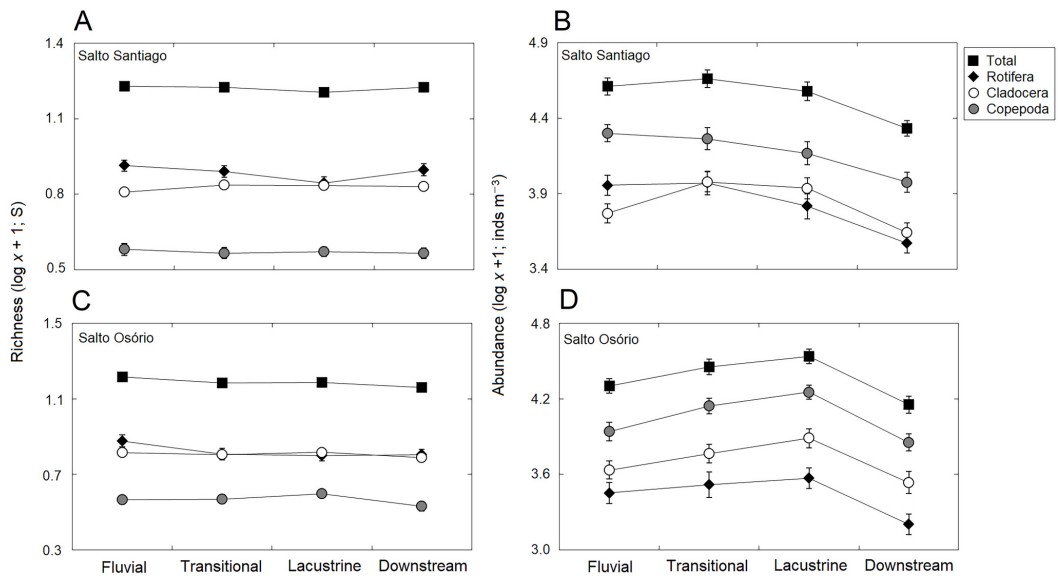

**Figure 6 Spatial variations in the richness and abundance of zooplankton.** Data shows the mean (±SE) richness and abundances of the total zooplankton, rotifers, cladocerans, and copepods in the (A, B) Salto Santiago and (C, D) Salto Osório reservoirs.

had greater abundance values upstream of the dam (fluvial-lacustrine zones), whereas the SO reservoir values were higher in the lacustrine zone compared to the higher flow environments (fluvial and downstream zones) (Fig. 6).

## Ordination of the zooplankton community

The NMDS ordination summarized the structure of the zooplankton communities and separated the years and periods for the SS reservoir and the years, periods, and environments for the SO reservoir (Fig. 7). The ANOSIM results also showed that the structure of the zooplankton communities in the two reservoirs were significantly different between years (see Table S2) and between periods. Finally, community structure only changed among the environments in the SO reservoir, where there was a significant difference ($P = 0.042$) between the lacustrine and downstream zone.

According to the SIMPER test, the species that most contributed to the differences in composition between the sampling years (2003–18) were *Polyarthra* sp. (11.0%), *C. cornuta* (10.6%), and *Conochilus* sp. (10.4%) in the SS reservoir; and *C. cornuta* (13.1%), *B. hagmanni* (9.2%), and *Ceriodaphnia silvestrii* (8.7%) were the most representative in the SO reservoir. The species that contributed the most to the differences in composition between the winter and summer periods were the same as for the sampling years, but with different contribution percentages. The species that contributed the most to the period differences were *C. cornuta* (11.3%), *Conochilus* sp. (10.4%) and *Polyarthra* sp. (10.1%) in the SS reservoir; and *C. cornuta* (13.5%), *B. hagmanni* (9.0%), and *C. silvestrii* (8.7%) in the SO reservoir (Table 4). It was possible to clearly discern temporal changes in zooplankton composition and the results showed that there was species substitution over time (Fig. 8).

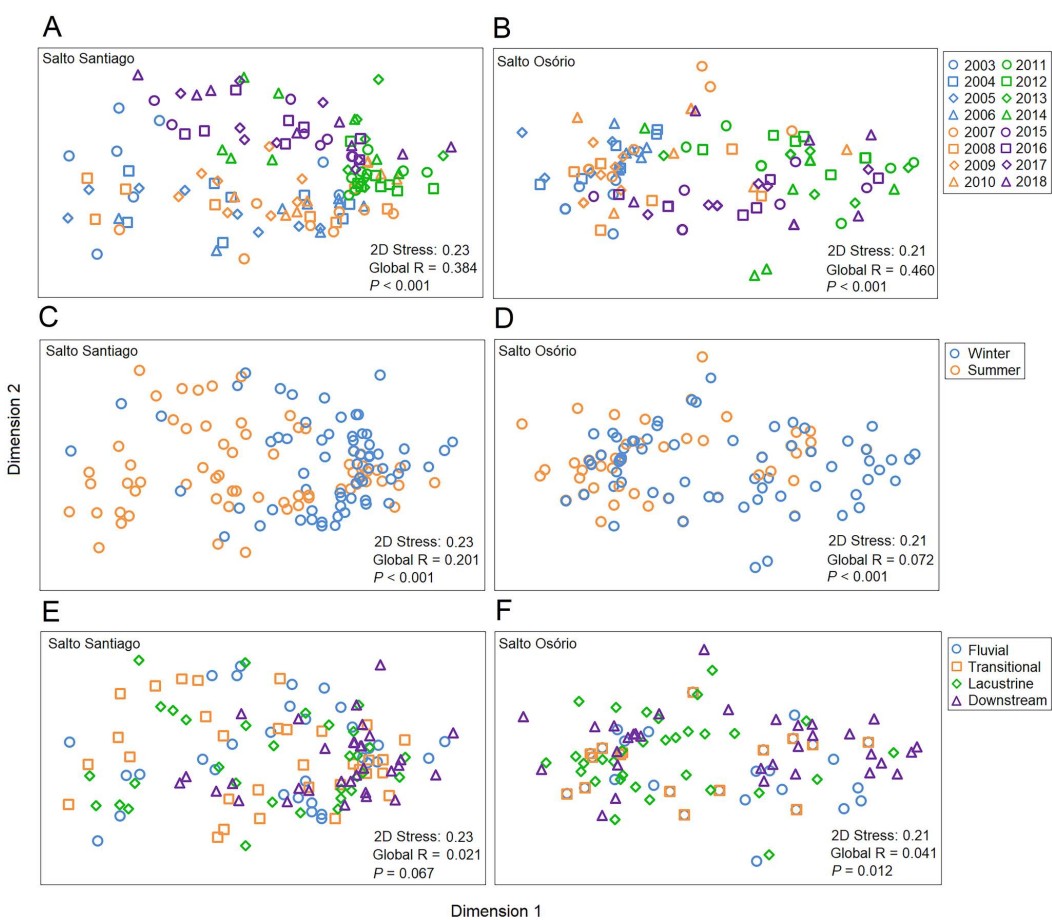

**Figure 7** **Ordination diagrams for the non-metric multidimensional scaling (NMDS) analysis.** Positions respective to the sampling year, period, and environment were used as symbol factors for the zooplankton communities in the (A, C, E) Salto Santiago and (B, D, F) Salto Osório reservoirs.

Finally, zooplankton composition differences among the environments only occurred in the SO reservoir. The species that contributed most to the differences were *C. cornuta* (13.1%), *B. hagmanni* (9.1%), and *C. silvestrii* (8.7%) (Table 4).

## Relationships between zooplankton community dynamics and the environmental variables

The first two CCA axes for the SS reservoir were significant ($P = 0.001$) and together explained 84.6% of the total variability in the zooplankton data. The first two axes of the CCA were also significant ($P = 0.001$) for the SO reservoir and together explained 73.8% of the total variability in the zooplankton data (Fig. 9). The first axes of the CCAs expressed differences between the years of study for the two reservoirs. Between the years 2003–12, some rotifers (e.g., *Euchlanis dilatata* and *Trichocerca cylindrica*) and microcrustaceans (*D. gessneri*, *B. hagmanni*, and *N. transitans*) were strongly associated with high electrical conductivity, $Z_{SD}$, WT, and the TP concentration. In contrast, the years 2013–18 were characterized by higher turbidity and higher Chl-*a*, TDS, and NT concentrations, which
**Table 4** **Similarity of percentages analysis (SIMPER) indicating zooplankton species contributions to composition among years, periods, and environments in the Salto Santiago and Salto Osório reservoirs.**

| Taxa | Salto Santiago | | | Salto Osório | | |
|---|---|---|---|---|---|---|
| | Year | Period | Environment | Year | Period | Environment |
| Rotifera | | | | | | |
| *Asplanchna sieboldii* | 8.0 | 7.5 | na | 2.2 | 2.2 | 2.3 |
| *Brachionus calyciflorus* | — | — | na | 1.5 | 1.5 | 1.5 |
| *Conochilus* sp. | 10.4 | 10.4 | na | 5.1 | 5.3 | 5.2 |
| *Euchlanis dilatata* | 2.2 | 2.3 | na | 2.6 | 2.7 | 2.6 |
| *Hexarthra* sp. | 3.7 | 4.3 | na | 2.8 | 3.0 | 2.9 |
| *Kellicottia bostoniensis* | 3.1 | 2.9 | na | 3.3 | 3.1 | 3.2 |
| *Polyarthra* sp. | 11.0 | 10.1 | na | 7.2 | 6.9 | 7.3 |
| *Synchaeta* sp. | 3.3 | 3.0 | na | 2.6 | 2.5 | 2.6 |
| *Trichocerca cylindrica* | 1.6 | 1.6 | na | 2.1 | 2.2 | 2.1 |
| Cladocera | | | | | | |
| *Bosmina hagmanni* | 9.1 | 8.9 | na | 9.2 | 9.0 | 9.1 |
| *Ceriodaphnia cornuta* | 10.6 | 11.3 | na | 13.1 | 13.5 | 13.1 |
| *Ceriodaphnia silvestrii* | 5.4 | 5.4 | na | 8.7 | 8.7 | 8.7 |
| *Daphnia gessneri* | 3.7 | 3.6 | na | 7.8 | 7.8 | 7.7 |
| *Diaphanosoma spinulosum* | 3.2 | 3.6 | na | 4.1 | 4.5 | 4.2 |
| *Moina minuta* | 4.6 | 4.9 | na | 6.9 | 7.0 | 6.8 |
| Copepoda | | | | | | |
| *Notodiaptomus henseni* | 3.4 | 3.5 | na | 5.0 | 4.8 | 5.1 |
| *Notodiaptomus transitans* | 4.8 | 5.0 | na | 4.5 | 4.3 | 4.4 |
| *Thermocyclops decipiens* | 1.7 | 1.6 | na | 2.2 | 2.1 | 2.2 |
| Average dissimilarity (%) | 72.7 | 74.7 | na | 72.5 | 72.4 | 71.8 |

**Notes.**

Only taxa that contributed the most to dissimilarity (average dissimilarity values/SD > 1) are presented. Values represent the percentage contribution of each species to group dissimilarity; na, no analysis.

were mainly associated with small rotifers (e.g., *Kellicottia bostoniensis*, *Conochilus* sp., and *Polyarthra* sp.).

The second axes of the CCAs were related to the winter and summer periods in the reservoirs. The WT and Chl-*a* and TDS concentrations were elevated in summer, which favored species such as *Asplanchna sieboldii*, *Conochilus* sp., *Moina minuta*, *Diaphanosoma spinulosum*, and *N. henseni*. In contrast, the higher $Z_{SD}$, and TP, and DO concentrations in winter favored some rotifer species (e.g., *K. bostoniensis* and *Polyarthra* sp.) and microcrustaceans (e.g., *Thermocyclops decipiens*, *D. gessneri*, and *B. hagmanni*) (Fig. 9).

# DISCUSSION

## Water characteristics of the two reservoirs

In the long-term, the rainfall patterns show a trend of low and/or absent seasonality for the study region. Consequently, there were few seasonal changes (except for DO, temperature, and TP) in the water characteristics. In addition, the lack of seasonality in the water properties may also be associated with the characteristics of the reservoirs, which

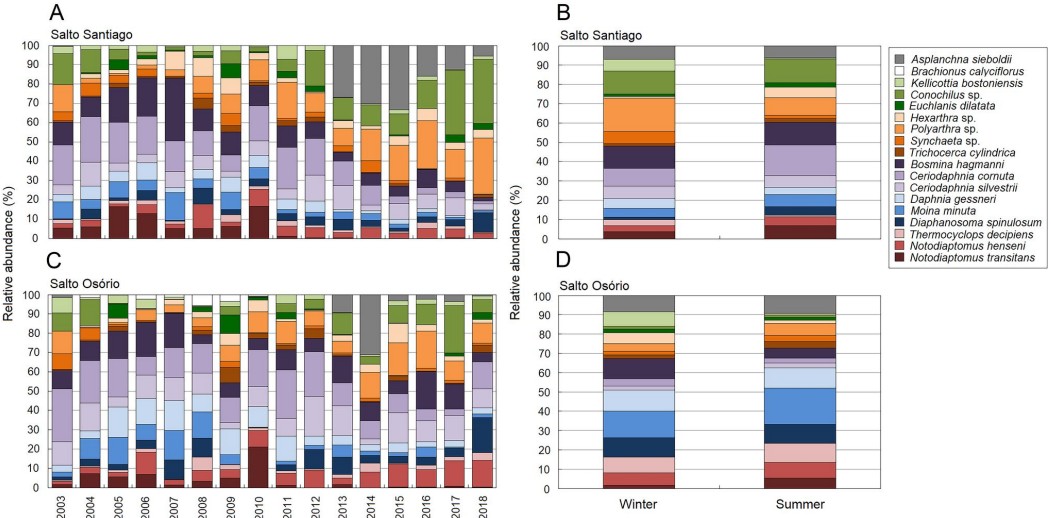

**Figure 8** **Zooplankton relative abundances for each year and period.** Percentage contribution of zooplankton species in the mean total abundance of zooplankton in the (A, B) Salto Santiago and (C, D) Salto Osório reservoirs. Only the species that the SIMPER test had shown contributed the most to community temporal variation are shown.

are manipulated according to their operational needs. Reservoirs temporarily accumulate rainwater that is gradually released. This process delays and mitigates flow peaks, which means that, in contrast to rivers, reservoirs do not follow a seasonal pattern (*Ravazzani et al., 2014*; *Ignatius & Rasmussen, 2016*). Even run-of-river reservoirs, such as SO, show large changes in their flow regime that are inconsistent with natural regimes (*Baumgartner, Baumgartner & Gomes, 2017*). Furthermore, the lack of seasonality in the SO reservoir may be closely linked to the operational needs of the upstream SS reservoir, which has a major influence on the flow regime.

The results showed that the inter-annual changes in rainfall and water temperature contributed to the large annual variability in the physical and chemical characteristics of the water. Inter-annual climatic phenomena, such as *El niño*, directly act on the amplitude of meteorological events, such as precipitation. This means that they can modify water properties, which may include changes in dissolved oxygen concentration, transparency, stratification, and primary productivity (*O'Reilly et al., 2003*; *Verburg, Hecky & King, 2003*; *Jankowski et al., 2006*; *Hampton et al., 2008*; *Marcé et al., 2010*), etc. Most previous studies related these modifications to changes in reservoir water flow, and the thermal balance and mixing dynamics of water bodies.

The high values for Chl-*a* and TSI in the storage reservoir may be linked to the long WRT and the large number of tributaries that drain into the SS reservoir, some of which drain urban areas, which means that they carry a large amount of nutrients. WRT is a determining factor in stratification processes and nutrient availability, promoting phytoplankton development as a consequence of higher nutrient concentrations and water column stability (*Londe et al., 2016*). In addition, the cascading effects of the reservoirs promote

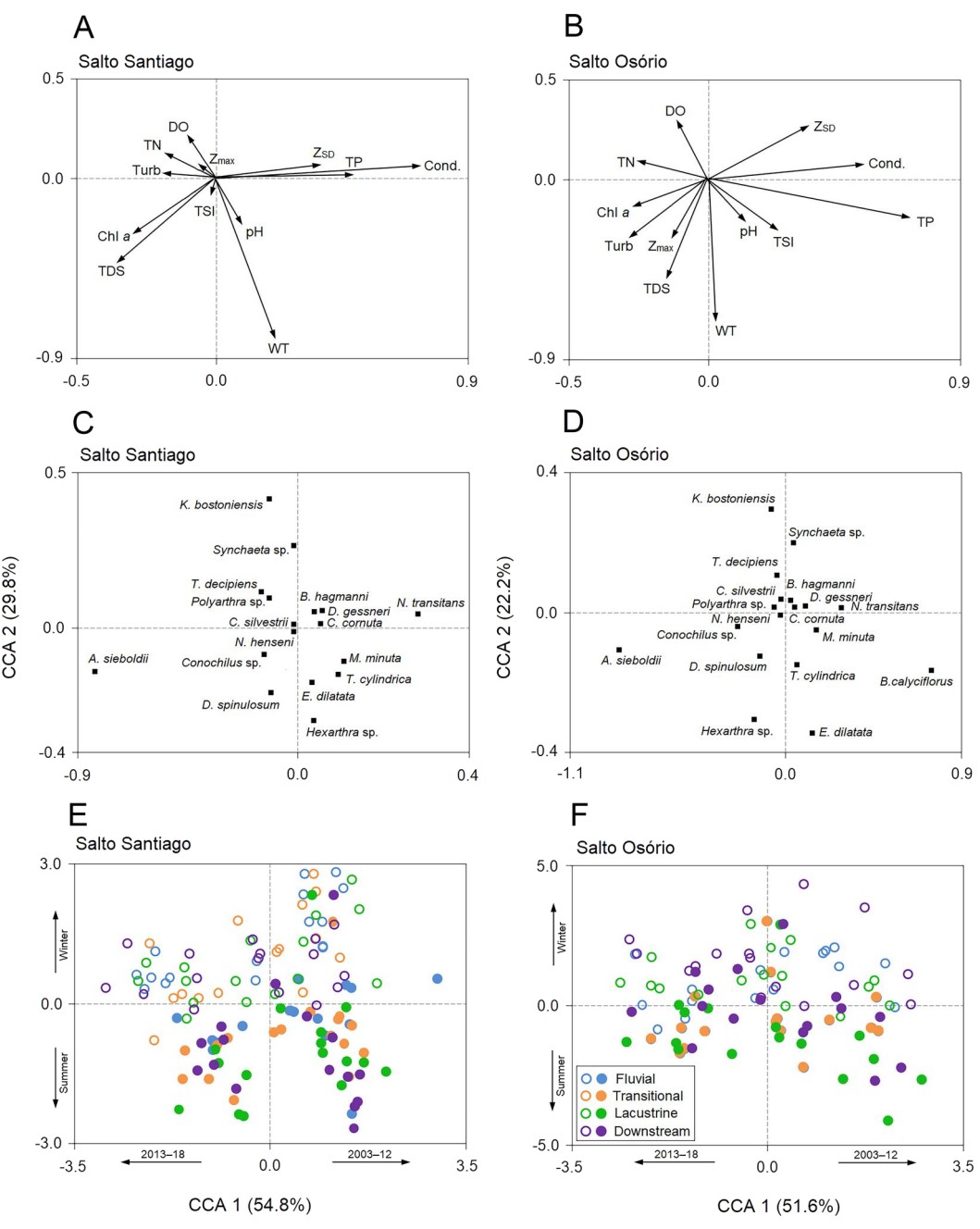

**Figure 9 Canonical correspondence analysis (CCA).** The CCA shows the relationships among the environmental variables, zooplankton species, and the environments of the (A, C, E) Salto Santiago and (B, D, F) Salto Osório reservoirs. Open circles (winter), closed circles (summer), chlorophyll $a$ (Chl $a$), dissolved oxygen (DO), maximum depth ($Z_{max}$), Secchi disk depth ($Z_{SD}$), electrical conductivity (Cond), total dissolved solids (TDS), total nitrogen (TN), total phosphorus (TP), turbidity (Turb), water temperature (WT), and trophic state index (TSI).

changes in the aquatic environment. The storage reservoir releases more transparent waters with low concentrations of suspended solids because most of the sediment is retained in the upstream compartments (*Ney, 1996*; *Padisák, Köhler & Hoeg, 1999*). This can lead to downstream oligotrophication. These changes in the longitudinal gradients of a reservoir cascade are predicted by the "cascading reservoir continuum concept" (*Barbosa et al., 1999*).

The differences in terms of size, morphometry, and the WRT between the two reservoirs also explain the extent to which tributaries influence reservoirs. The modifications due to tributary entrances in run-of-river reservoirs are smaller and are associated with semi-lotic conditions (*Nogueira, Perbiche-Neves & Naliato, 2012*). Furthermore, there is not enough time or distance for sediment loads to be deposited on the reservoir bed, and the simple SO reservoir morphometry does not favor the particle deposition process (*Perbiche-Neves & Nogueira, 2010*). The opposite situation occurs in SS reservoir, due to its dendritic shape and long longitudinal axis, which allows particle sedimentation and a longitudinal increase in transparency. This characteristic corroborates the foreseeable modifications towards the dam zone (*Thornton, Kimmel & Payne, 1990*; *Pinto-Coelho et al., 2006*; *Soares et al., 2012*). These are due to reduced water velocity, and increased depth and WRT. The WRT is a fundamental variable for reservoir ecology, which interferes with physical, chemical, and biological characteristics, and depends on the interaction between different factors, such as precipitation, flow, evaporation, and infiltration (*Nogueira, Perbiche-Neves & Naliato, 2012*; *Beaver et al., 2013*).

## Zooplankton community structure and temporal variation in the two reservoirs

Simultaneous extreme weather events, such as heavy rainfall and flooding, and severe droughts, have been affecting South America in recent years (*Rocha et al., 2019*). These opposite extreme events may have affected both reservoirs between 2003–18, leading to significant changes in nutrient loading with complex effects on zooplankton structuring. The zooplankton structure (richness, abundance, and composition) underwent important changes, some of which coincided with increased nutrient loading and trophic state changes in the reservoirs. The decrease in the abundance of large zooplankton filter-feeding species, such as large rotifers (e.g., *Brachionus calyciflorus* and *Euchlanis dilatata*), cladocerans (e.g., *D. gessneri*) and calanoid copepods (*N. transitans*), coincided with an increase in the abundance of other species, such as the rotifer *A. sieboldii*, small rotifers with generalist characteristics (e.g., *Conochilus* sp. and *Polyarthra* sp.,) and small cladocerans (e.g., *B. hagmanni* and *Ceriodaphnia spp.*).

The use of zooplankton species as biological indicators can provide important information about current and past processes, such as changes in biological relationships and in the physical and chemical properties of water (*Perbiche-Neves et al., 2019*). For example, the cladoceran *D. gessneri* and calanoid copepods (especially *N. transitans*) were abundant in the years with higher water transparency and lower concentrations of Chl-*a* and TDS (2003–11). The filter-feeding activity of these microcrustaceans can have a great influence on Secchi transparency values (*García-Chicote, Armengol & Rojo, 2018*),

as they are able to eliminate nanoplankton in a large area of the water. Concentrations of small particles (e.g., nanoplankton) affect Secchi depth values to a greater extent than those of larger particles (*Horn & Horn, 1995*), resulting in greater Secchi depths in the years when there were high abundances of daphinids and calanoid copepods. However, the predominance of young copepod forms (nauplii and copepodites) and small cladocerans (*B. hagmanni* and *Ceriodaphnia spp.*), which were mainly observed between 2013–2014, indicate that mesotrophic conditions prevailed in both reservoirs. Environments with higher concentrations of detritus and nutrients (mesotrophic) favor the growth of bacteria and protozoa, an important source of food for small filter-feeders such as nauplii, rotifers, and small cladocerans (*Brito, Maia-Barbosa & Pinto-Coelho, 2011*). In addition, nutrient enrichment in reservoirs may indicate the appearance of cyanobacteria in the phytoplankton. Reports of cyanobacterial blooms in oligo-mesotrophic reservoirs are not uncommon (*Brito, Maia-Barbosa & Pinto-Coelho, 2011*; *Su et al., 2019*). Some previous studies in the SS and SO reservoirs have also recorded large abundances of cyanobacteria in the phytoplankton (*Adloff et al., 2018*; *Bortolini et al., 2018*). The filaments or mucilages of these algae directly interfere with the filtration apparatuses of large cladocerans and calanoid copepods, which leads to the decline and replacement of these populations by small rotifers and cladocerans (*Sendacz, Caleffi & Santos-Soares, 2006*; *Eskinazi-Sant'Anna et al., 2013*; *Perbiche-Neves et al., 2016*). In addition, a greater abundance of young copepod stages than adults may indicate an adaptive strategy to compensate for the high mortality of young stages (*Lansac-Tôha, Bonecker & Velho, 2005*).

In both reservoirs, rotifers were the organisms that most contributed to the total species richness of the zooplankton. The high diversity of this group in reservoirs has been a recurring pattern in Brazil and is mainly attributed to the opportunistic characteristics of this group. Rotifers occupy a wide range of water bodies, have a wide food spectrum (from bacteria to filamentous algae), have a high population turnover rate, and are more diverse in the littoral areas of oligo-mesotrophic water bodies (*Segers, 2008*). Some studies have associated the presence of certain rotifer species with the trophic conditions of the environments in which they are found. For example, *Nogueira (2001)* and *Sampaio et al. (2002)* related the presence of the rotifers *Polyarthra* sp. and *Conochilus* sp. with the oligotrophic conditions of Paranapanema River reservoirs. In this study, these species were predominant in the reservoirs mainly during the last few years (after 2014), when the waters were more oligotrophic. In addition, the increase in the abundance of the rotifer *A. sieboldii* in these years, especially in the SS reservoir, coincided with an increase in the abundance of these small rotifers, and in the concentrations of Chl-*a* and TDS. The genus *Asplanchna* are considered voracious predators, usually cannibals, and can feed on almost all species of rotifers, in addition to consuming phytoplankton and detritus (*Gilbert, 1980*; *Thatcher, Davis & Gardner, 1993*). Authors such as *Choi et al., (2015)* observed that the preferred prey of *Asplanchna* sp. is pelagic rotifers of the genera *Polyarthra* and *Keratella*. Thus, the inter-annual variations in trophic conditions also played an important role in increasing *A. sieboldii* populations, owing to the greater availability of prey. In this context, knowledge of zooplankton fluctuations, both seasonal and inter-annual, is essential to assess the effects of climate change on nutrient transfer processes in aquatic food networks (*Caroni & Irvine,*
*2010*; *García-Chicote, Armengol & Rojo, 2018*). Therefore, from an ecological point of view, zooplankton plays an essential role within the trophic networks of lakes and reservoirs, as it has high value as an indicator that cannot be replaced by phytoplankton and fish studies (*Jeppesen et al., 2011*).

The results for seasonal changes in the attributes of zooplankton communities showed that only species richness did not change. The low seasonal fluctuations in rainfall and reservoir flow patterns probably contributed to the richness homogeneity between the periods. In contrast, there were seasonal changes in abundance and species composition. The fall in abundance values during winter was mainly linked to the decrease in water temperature. Changes in temperature can trigger seasonal processes that influence zooplankton directly (e.g., metabolic rates, body size, egg production) and indirectly (e.g., phytoplankton structure) (*Hart & Bychek, 2011*; *Brito, Maia-Barbosa & Pinto-Coelho, 2016*). Other studies have also observed a decrease in zooplankton abundance in tropical reservoirs during periods with lower temperatures (*Sendacz, Caleffi & Santos-Soares, 2006*; *Tundisi, Matsumura-Tundisi & Abe, 2008*; *Sartori et al., 2009*). In temperate lakes, there is a noticeable decrease in zooplankton abundance during winter and an increase in the abundance of new species during warmer periods (spring and summer) (*Hessen et al., 2006*; *Beaver et al., 2013*). It is probable that harsh winters in southern Brazil are an important regulator of zooplankton community structure.

The rotifers *Synchaeta* sp., *Polyarthra* sp., and *K. bostoniensis* (an invasive species originating in North America) were predominant during the winter periods between 2013–18, and had a strong relationship with decreasing water temperature and increased turbidity in the SS reservoir, and the TN concentrations in both reservoirs. Some rotifers exhibit a moderate tolerance to temperature variation and are common in nutrient-rich environments where they efficiently consume solid suspended particles and colloids from bacteria that break down organic matter (*Sluss, Cobbs & Thorp, 2008*; *Balkić, Ternjej & Špoljar, 2016*). However, during the summer periods of the same years, the copepod *N. henseni* and the cladoceran *D. spinulosum* were more abundant in both reservoirs. The evaluated data indicated that, besides the temperature, the higher food availability (Chl-*a* and TDS) was also responsible for the increase in these organisms. Therefore, higher summer temperatures may have favored higher phytoplankton productivity, which led to favorable conditions for the development of these populations. The cyclopoid copepod *T. decipiens* was common in the zooplankton during the winter periods. Cyclopoid copepods preferably consume algae and, additionally, other invertebrates present in zooplankton or in the benthic and littoral areas (*Dussart & Defaye, 1995*; *Velho, Lansac-Tôha & Bonecker, 2005*; *Perbiche-Neves et al., 2016*). The lower winter temperatures did not limit the development of *T. decipiens* because it has a wide food spectrum.

During the winter periods between the years 2003–12, there was also a clear increase in the populations of microcrustaceans *B. hagmanni*, *D. gessneri*, *C. cornuta*, and *N. transitans* in both reservoirs. The interactions among factors such as decreased transparency, competition for resources, and fish predation pressure may have limited the development of these winter favored populations in the summer. Like phytoplankton, zooplankton have characteristic seasonal cycles that are strongly linked to temperature seasonality,

hydrology, food availability, and predation pressure, and variations in these factors can affect the populations of these organisms (*Havens et al., 2009*; *Beaver et al., 2018*; *Hu et al., 2019*). However, previous studies (e.g., *Havens, 2002*; *Pinto-Coelho et al., 2005)* and our results indicate that there appears to be a greater correlation between microcrustaceans and TP concentrations than between microcrustaceans and Chl-*a* in the SS reservoir. Sometimes the primary productivity of algae is lower than the secondary productivity of zooplankton. Similarly, primary productivity by algae may not cover the high rates of assimilation by zooplankton. At this point, the zooplankton probably begin to consume suspended detritus or bacterioplankton. When this happens, TP may be a better indicator of food availability and better able to explain zooplankton variability than Chl-*a* (*Havens, 2002*; *Pinto-Coelho et al., 2005*; *Sluss, Cobbs & Thorp, 2008*).

Studies on the functioning and structure of zooplankton communities in reservoir ecosystems offer opportunities to investigate patterns of response to cyclical variations and episodic disturbances. An understanding of planktonic dynamics in reservoirs can also be useful to assess the resilience of this type of ecosystem over relatively short periods, which can present profound changes in the intra- and inter-reservoir limnological conditions arranged in the cascade. This dynamic is generated by short-term variations in the flow regime, water level, and WRT and by interactions with other aquatic and terrestrial organisms present in the hydrographic basin (*Barbosa et al., 1999*; *Nogueira, 2001*).

## Zooplankton differences between the two reservoir environments

The characterization of the zooplankton community and its spatial distribution provide important data for the study of reservoirs, facilitating a broader understanding of these environments and allowing for adequate monitoring and management (*Nogueira, 2001*; *Bernot et al., 2004*). The zooplankton richness, composition, and abundance results showed that there were differences between the reservoirs. The higher richness and abundance in SS were probably related to the higher phytoplankton productivity, which was indicated by the higher Chl-*a* and TSI values. Some previous studies have suggested that there was also a strong link between the zooplankton community and the functional characteristics of the reservoirs (*Bini et al., 2008*; *Nogueira, Oliveira & Britto, 2008*). They suggested that the degree of compartmentalization, stability of the different water masses, and the presence of vertical stratification had crucial effects on zooplankton community structure. For example, they can lead to an increase in the richness and abundance values (*Beaver et al., 2015*). Furthermore, a low WRT has been shown to be an important factor that limits zooplankton abundances in many run-of-river reservoirs (e.g., *Perbiche-Neves & Nogueira, 2013*; *Beaver et al., 2013*).

The longitudinal total richness of the zooplankton was not the same in the two reservoirs. Only the SO reservoir showed heterogeneity for this attribute, with a higher number of species in the fluvial zone compared to downstream. The higher number of species in the fluvial zone of the SO reservoir may be related to species increments from the SS reservoir, especially planktonic species that are derived from the lacustrine zone. However, the homogeneous distribution of richness in the SS reservoir may be related to its improved stability. Only the rotifers exhibited a longitudinal gradient for richness, with a greater

number of species in the fluvial and transitional zones. The greater number of species in these environments may be related to the higher current velocity and the lower depth compared to the lacustrine zone. These factors would increase transportation to the water column of species from the littoral and benthic areas (*Serafim-Jr et al., 2016*). Some studies have also reported greater rotifer richness in the fluvial and transitional zones. They attributed this to the incorporation of non-planktonic and planktonic species from lateral tributaries and reservoirs located upstream of the channel, respectively (*Marzolf, 1990*; *Velho, Lansac-Tôha & Bonecker, 2005*; *Czerniawski & Kowalska-Góralska, 2018*).

The abundance values also showed a longitudinal gradient in both reservoirs, with higher values in environments with lentic characteristics. The largest zooplankton abundances, especially microcrustaceans, occurred in the transitional and lacustrine environments, and this may be related to the lower flow velocity and turbidity, and higher values for water transparency, dissolved oxygen, and primary productivity in these compartments (*Hart, 2004*; *Sartori et al., 2009*; *Brito, Maia-Barbosa & Pinto-Coelho, 2016*). When flow velocity is high, zooplankton population growth can be inhibited, even when food resource levels are high (*Beaver et al., 2015*). The greater depth in these compartments may also benefit the development of species with vertical migration behavior, such as microcrustaceans (*Perbiche-Neves & Nogueira, 2013*).

Longitudinal changes in species composition were only observed in the SO reservoir where the main differences were between the lacustrine and downstream zones. The WRT differences influence the limnological characteristics and dynamics of aquatic communities within reservoirs and in areas downstream of the dams (*Ferrareze, Casatti & Nogueira, 2014*). Therefore, the flow velocity in the channel below the two types of reservoirs will also differ and will differentially influence downstream species composition. The SIMPER test results showed that the cladoceran species, especially bosminids and daphinids, contributed most to differences among the environments. These microcrustaceans have planktonic characteristics and do not adapt well to the unstable conditions present in lotic environments, such as downstream of the SO reservoir. They mainly prefer places with slow or standing water (*Viroux, 2002*). Several studies have suggested that there is a larger decrease in cladoceran populations in lotic environments than small rotifer populations, which is probably caused by high mortality rates due to turbulence, and limitations to growth and reproduction (*Baranyi et al., 2002*; *Sluss, Cobbs & Thorp, 2008*). In addition, the type of filtration feeding has been cited as one of the factors that make these cladoceran species so successful in lentic environments.

## CONCLUSIONS

The WRT seems to play an important role in zooplankton community structure in this cascade system and is directly linked to the functional characteristics of the reservoirs. The greater stability of the water masses and higher primary productivity in the storage reservoir (SS) provide a more suitable environment for the development of zooplankton populations, mainly microcrustaceans. In contrast, the location (downstream) and shorter WRT of the run-of-river reservoir (SO) negatively affects species richness and zooplankton

abundance, because they decrease trophic conditions and increase water flow, respectively. The SS reservoir (upstream) influences the limnological and biological characterization of the SO reservoir (downstream), mainly owing to the release of more oligotrophic waters and the high export rates of zooplankton downstream, making the downstream reservoir dependent on the water level management actions of the upstream reservoir. In contrast, the SS reservoir is more dependent on its intrinsic characteristics (e.g., its longitudinal axis and depth), which provide for the formation of environmental gradients that act directly on the zooplankton structure. Thus, these differences between the reservoirs result in changes both in water characteristics and in the structure of the zooplankton community throughout the cascade system. Annual and seasonal changes in water quality and water flow displayed equally important effects on the temporal variation in the zooplankton structures in both reservoirs, and it was possible to observe species substitutions over time. Despite their important ecological role in aquatic systems, few environmental policies consider zooplankton as a tool to strengthen strategies for managing and maintaining the biological diversity of these environments. These results support the utilization of these organisms as a useful tool to improve our understanding of changes in water quality and the ecosystem processes involved in these changes.

## ACKNOWLEDGEMENTS

We would like to thank the GERPEL (Grupo de Pesquisas em Recursos Pesqueiros e Limnologia), INEO (Instituto Neotropical de Pesquisas Ambientais), and UNIOESTE (Universidade Estadual do Oeste do Paraná) for providing technical support. We are also grateful to the anonymous reviewers, whose detailed comments and constructive suggestions improved the quality of the manuscript.

### Funding

This work was supported by the CAPES (Coordenação de Aperfeiçoamento de Pessoal de Nível Superior) through a post-doctorate fellowship granted to Pablo Henrique dos Santos Picapedra. The funders had no role in study design, data collection and analysis, decision to publish, or preparation of the manuscript.

### Grant Disclosures

The following grant information was disclosed by the authors:
CAPES (Coordenação de Aperfeiçoamento de Pessoal de Nível Superior).

### Competing Interests

The authors declare there are no competing interests.

### Author Contributions

- Pablo H.S. Picapedra and Cleomar Fernandes conceived and designed the experiments, performed the experiments, analyzed the data, prepared figures and/or tables, authored or reviewed drafts of the paper, and approved the final draft.

- Juliana Taborda conceived and designed the experiments, performed the experiments, analyzed the data, authored or reviewed drafts of the paper, and approved the final draft.
- Gilmar Baumgartner and Paulo V. Sanches conceived and designed the experiments, authored or reviewed drafts of the paper, and approved the final draft.

## Data Availability

The raw data of the environmental variables and zooplankton density are available in the Supplementary Files.

## Supplemental Information

Supplemental information for this article can be found online at http://dx.doi.org/10.7717/peerj.8979#supplemental-information.

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
