# Peer review of "A long-term study on zooplankton in two contrasting cascade reservoirs (Iguaçu River, Brazil): effects of inter-annual, seasonal, and environmental factors"

_PeerJ, doi:10.7717/peerj.8979_

## Round 0.1 · original submission · Major Revisions

I agree with both reviewers that this is an interesting paper.
However, I also agree with Reviewer #2 that the paper needs major revisions. I strongly recommend that the authors address carefully all points made by Reviewer #2, as well as the main criticism of Reviewer #1 ("The authors didn’t answer if the Salto Santiago and Salto Osório reservoirs are independent systems. That was the second hypothesis stated in the manuscript introduction.").

·

Basic reporting

There are two instances are missing a bibliographic reference to endorse the authors arguments:
- In the Introduction section, lines 85 and 86, where the authors indicate say “Fast responses to climate change are expected in systems where lotic conditions predominate, unlike reservoirs operated as lentic environments.”
- In the Discussion section, line 454 to 457, when the indicate “a. The primary productivity of algae is often lower than secondary productivity by zooplankton. Primary productivity by algae may not cover the high rates of assimilation by zooplankton. At this point, the zooplankton probably begin to consume suspended detritus or bacterioplankton. When this happens, TP could be a better indicator of food availability.”.

Finally, in Table 2 and the Supplemental data must be revised. The numbers are displayed with commas, where they should be stops. Ex: "3,313" should be changed to "3.313".

Experimental design

There are two instances that should be reviewed in the Materials & Methods section:
First, in line 144, the brand and model of the “portable digital devices” that the authors used is missing.
Secound, in line 202, the authors should explain, in the Materials & Methods section, why they transformed all data to log (x + 1).

Validity of the findings

The authors didn’t answer if the Salto Santiago and Salto Osório reservoirs are independent systems. That was the second hypothesis stated in the manuscript introduction. I think that it should be clearer in the conclusions.

Additional comments

The manuscript is very interesting and has been conducted with extreme dedication, evident in an impressive sample effort.
Owing to the fact that dam environments are not significantly influenced by seasonality, according to the authors, a long-term study, like the 16 years carried out in the reviewed study, may indicate how the zooplankton community varies between inter-annual climatic phenomena. Results such as these may greatly enrich many studies discussions, such as those concerning aquatic environments susceptible to eutrophication and toxicity processes caused by urban growth, species substitutions over time, as well as research concerning climate change effects on closed aquatic ecosystems.

Reviewer 2 ·

Basic reporting

The manuscript titled “A long-term study on zooplankton in two contrasting cascade reservoirs (Iguaçu Ricer, Brazil): Effects of inter-annual, seasonal and environmental factors” is an interesting read. The paper focuses on a long-term compilation data from two reservoirs. Studies focusing in the understanding of zooplankton and environmental metadata are very important to possibility the variations of man-made lakes in a large temporal scale. The authors had a significant sample size which allowed them to perform a good statistical analysis, comparing two different reservoirs with different hydrodynamics and environmental conditions. Therefore, the authors could improve the discussion of the hypotheses and do provide evidence for its discussion and conclusion. The main hypotheses of the study should be presented and discussed more objectively. The Discussion section would need to address the aspects that support the changes in the zooplankton community in a more concise manner and supported by the real results obtained. An English revision is also suggested. Therefore, I would recommend major revisions to the manuscript that would make it suitable for publication.

Experimental design

No comment

Validity of the findings

No commet

Additional comments

General comments:

1. The paper would greatly benefit from rewriting the Introduction using more relevant and recent references and more ecological theory could be included. The authors could also show how their work builds on previous research and how they reached their conclusions. It should be clear where and how this research fits in with this specific field and what knowledge gaps are attempted to be filled.
2. Some results should be reanalyzed and rewritten. It is not clear, for example, if the sampling of the environmental variables was conducted at surface or in vertical profiles of the water, so it would be important to provide information on this.
3. The authors should emphasize that zooplankton studies may be useful in the prediction of long-term changes in dynamic aquatic ecosystems such as reservoirs and used in water quality assessment initiatives. They only considered this aspect in the final paragraph of the conclusions.
4. The temporal aspects of the changes in zooplankton community are attributed by the authors to the eutrophication, but the trophic state index indicates a predominance of meso/oligotrophic conditions in the last 5 years in the reservoirs (mainly in lacustrine and downstream regions – supported by the ANOSIM test). The pattern of change in the zooplankton community should also be discussed considering this aspect.

Specific remarks:

L105 There is a contradiction here. The authors made the statement that “zooplankton richness, composition and abundance were negatively affected by ‘decreased trophic conditions”, but in the Discussion the eutrophication is considered the main trigger to the changes in the zooplankton structure.
L116 – “and is caused” … should be rephrased
L143 – Limnological parameters were obtained at surface of the water?
L151 – Data of water flow provided by ANA was measured in one area at the downstream of the reservoir? Details about the collection of these data should be provided. Precipitation values were also obtained from ANA through a local climatological station?
L 155 – suggestion: “The zooplankton samples (720 in total)”
L167 – Rephrased the last line “there were 720 samples in total”
L211 – “precipitation had no direct effect on water flow regulation”. How is this result supported by statistical analysis? This phrase is not clear, should be rephrased
L213 – Please see Fig 2. In 2005 and 2017 there were high peaks oh cumulative rainfall. Recheck the results.
L218 – “revealed that a small amount of eutrophication had occurred”… This phrase makes no sense. The trophic condition of the reservoirs is a dynamic, successional event. Please rephrase
L346 – It should not also be related to the highest WRT?
L375 – Is this observation supported by the obtained results? The most abundant species of rotifers registered in the reservoir are planktonic, except Euchlanis.
L395 – How could the process of oligotrophication observed in the reservoirs explain the results?
L477 – the authors are considering again the aspect of fluvial transport of zooplankton, already considered in line 375. I suggest considering the aspects of global influence of environmental features of the reservoirs to discuss this aspect of the zooplankton community.

Annotated reviews are not available for download in order to protect the identity of reviewers who chose to remain anonymous.

---

## Round 0.2 · accepted · Accept

Both reviewers considered that the revisions addressed all prior concerns. The manuscript is considered a strong "contribution to the ecology of tropical reservoirs, with emphasis on the potential of the zooplankton as bioindicators of environmental conditions."

·

Basic reporting

I believe that all revision concerns have been carried out.

Experimental design

I believe that all revision concerns have been carried out.

Validity of the findings

I believe that all revision concerns have been carried out.

Additional comments

I believe that all revision concerns have been carried out.

Reviewer 2 ·

Basic reporting

The authors of the manuscript made an appropriate correction to the text and clarified all the reviewer questions. The results are supported by appropriate statistical analysis.

Experimental design

Sampling design is appropriate for the main questions of the study. The authors clarified the review doubts about zooplankton and environmental data sampling. The manuscript is an excellent contribution to the ecology of tropical reservoirs, with emphasis on the potential of the zooplankton as bioindicators of environmental conditions.

Validity of the findings

The conclusions of the manuscript are linked to the main question and support the results.

Additional comments

The authors of the manuscript made an appropriate correction to the text and clarified all the reviewer questions. The results are supported by appropriate statistical analysis. The manuscript is an excellent contribution to the ecology of tropical reservoirs, with emphasis on the potential of the zooplankton as bioindicators of environmental conditions. In this way, I consider that the manuscript could be accepted for publication and would be an interesting theme for the Peer J readership.